# Real-Time Classification of Motor Imagery Using Dynamic Window-Level Granger Causality Analysis of fMRI Data

**DOI:** 10.3390/brainsci13101406

**Published:** 2023-10-01

**Authors:** Tianyuan Liu, Bao Li, Chi Zhang, Panpan Chen, Weichen Zhao, Bin Yan

**Affiliations:** Henan Key Laboratory of Imaging and Intelligent Processing, PLA Strategic Support Force Information Engineering University, Zhengzhou 450001, China; tytauceti@hotmail.com (T.L.);

**Keywords:** brain–computer interaction (BCI), real-time functional magnetic resonance imaging (rt-fMRI), motor imagery (MI), Dynamic Window-level Granger Causality (DWGC), Support Vector Machine (SVM)

## Abstract

This article presents a method for extracting neural signal features to identify the imagination of left- and right-hand grasping movements. A functional magnetic resonance imaging (fMRI) experiment is employed to identify four brain regions with significant activations during motor imagery (MI) and the effective connections between these regions of interest (ROIs) were calculated using Dynamic Window-level Granger Causality (DWGC). Then, a real-time fMRI (rt-fMRI) classification system for left- and right-hand MI is developed using the Open-NFT platform. We conducted data acquisition and processing on three subjects, and all of whom were recruited from a local college. As a result, the maximum accuracy of using Support Vector Machine (SVM) classifier on real-time three-class classification (rest, left hand, and right hand) with effective connections is 69.3%. And it is 3% higher than that of traditional multivoxel pattern classification analysis on average. Moreover, it significantly improves classification accuracy during the initial stage of MI tasks while reducing the latency effects in real-time decoding. The study suggests that the effective connections obtained through the DWGC method serve as valuable features for real-time decoding of MI using fMRI. Moreover, they exhibit higher sensitivity to changes in brain states. This research offers theoretical support and technical guidance for extracting neural signal features in the context of fMRI-based studies.

## 1. Introduction

The neural control system of brain–computer interaction (BCI) investigates the perception and neural decoding of multi-modal fusion across different time scales. Optimizing automatic recognition of neural states from feature models at three levels can enhance the overall performance of recognition algorithms in terms of accuracy, specificity, and model flexibility [1,2,3,4,5]. In BCI rehabilitation therapy, a therapeutic approach that identifies the stroke patient’s grasping motor intentions of both hands and provides feedback to the patient is effective [6]. At present, the limited availability of unilateral motor imagery (MI) rehabilitation actions [7] and the poor recognition ability [8] hinder the development of MI-BCI rehabilitation, posing an unresolved challenge [9].

MI, as a very important active brain–computer interface paradigm, is used to recognize the brain’s motor intention for limbs [10]. It can induce specific responses in the sensorimotor cortex without requiring external stimulation or obvious motor output. Researchers aim to enhance the MI capabilities through various approaches, and they often prioritize task variety over considering the impact of response sensitivity on training outcomes [11,12]. Meanwhile, significant changes in blood oxygen concentrations occur in different regions of the brain when patients perform tasks related to motor execution (ME) or MI, which can be recorded with fMRI. However, this approach is subject to decoding delay due to the delay effect of blood oxygen concentration changes. Improving response speed and classification accuracy can provide better feedback training effects for patients.

Several studies have shown that rt-fMRI MI training has the potential to induce changes in the functional connectivity between various brain regions. Thus, investigating functional connectivity as a characteristic of MI presents promising research opportunities. Currently, the analysis of neural signals in multiple brain regions extensively employs effective connectivity analysis based on Granger causality. Traditional Granger causality methods primarily rely on linear autoregressive models to process long-term time series [13]. In contrast, the establishment of a Dynamic Window-level Granger Causality (DWGC) model enables the extraction of features at the local brain network level, while also considering the real-time requirements of brain–computer interface (BCI) systems, thereby achieving a faster response speed [14].

In this study, we utilized a combination of MI preparation and MI experiments to identify the brain regions implicated in left- and right-hand MI. Subsequently, relevant seed nodes were extracted, and DWGC analysis was employed to determine the short-term effective connectivity between these seed nodes. Furthermore, we developed MI features based on the average BOLD activation values within the corresponding brain regions. To enable real-time classification of left- and right-hand MI, the Open-NFT open-source package [15] and SVM algorithm were integrated. The SVM algorithm is a commonly used algorithm in machine learning, which can solve complex nonlinear classification problems [16]. As a result, we successfully achieved real-time decoding of the MI for both hands. Notably, this method demonstrated an average classification accuracy of 62% in the three-classification task, surpassing the performance of traditional BOLD activation values. Additionally, it significantly improved the speed of decoding response.

## 2. Materials and Methods

This section describes the methods and steps involved in data processing and describes two experiments in detail. Additionally, it explains the process of identifying the regions of interest (ROIs) and how to calculate effective connectivity between target regions.

### 2.1. Ethics Statement

The experimental protocols conducted in this study were approved by the Ethics Committee of the Henan Provincial People’s Hospital (No. 2022-169). All research procedures were conducted in accordance with the relevant guidelines and regulations. Informed consent was obtained from all participants prior to their inclusion in the study, and they voluntarily signed a consent form indicating their willingness to participate in the experiment.

### 2.2. Subject

The subjects of this study were six right-handed adult males aged 20–25 years without any neurological disorders. Based on their overall physical and mental well-being, all participants had no history of mental illness and had never taken any psychotropic medication. And all subjects had Chinese as their native language and had normal or corrected-to-normal vision. All subjects were familiar with the experimental procedures in advance and had abundant experience with fMRI experiments.

### 2.3. Apparatus

All computer-controlled stimuli were programmed in Open-NFT and presented through a BOLDscreen monitor with 4K resolution (1920 × 1080, 120 Hz), with size of 89 cm × 50 cm and viewing distance of 168 cm. The stimuli were presented using a monitor positioned above the subjects’ heads. Subjects observed the content on the monitor through a head coil-mounted 45-degree mirror.

### 2.4. Experimental Procedures

This section provides a detailed description of the experimental design. The study is divided into four distinct steps. The initial step involves collecting brain data during left- and right-hand MI tasks. The second step involves processing the data to identify the brain regions associated with motor imagery activity. In the third step, features are extracted using DWGC. Finally, the extracted features are inputted into the classifier for training purposes. This process is described briefly in Figure 1.

### 2.5. Stimuli and Experiment

In order to persuade the subjects to complete the experimental tasks according to the requirements, vision stimuli were devised. The stimuli consisted of three types of cues: left-hand motion imagination cue, right-hand motion imagination cue, and rest cue. These cues corresponded to the subjects’ motion state during the experiment.

The pre-experiment was designed to locate the ROIs in the subjects’ brains. The ROIs were crucial for identifying the activated brain areas during left- and right-hand preparation, as well as motor imagery. The experimental paradigm was designed as follows: each subject performed four tasks in sequence: left-hand grasping motor imagination (MIL), right-hand grasping motor imagination (MIR), left-hand grasping motor preparation (MPL), and right-hand grasping motor preparation (MPR) (Figure 2C). Each task is completed under the stimuli symbol which lasted for 10 s. The tasks alternated continuously throughout the experiment, with a 20-s rest period in between. The total duration of the experiment was 300 s (Figure 2A).

The test-experiment gathered brain data during left- and right-hand motor imagination, processed the data in real time, and classified the corresponding motor imagination states. The subjects initially rested for 20 s, and then prompted to randomly imagine left- and right-hand movements, with each motor imagination lasting for 10 s, and the time of each motor was 10 s, followed by another 20-s rest period (Figure 2B).

### 2.6. Data Acquisition

All experimental data were collected using SIEMENS MAGNETOM Pria 3.0 T magnetic resonance scanner and 64-channel head coil, see Table 1 for specific scanning parameters.

### 2.7. Data Preprocessing

SPM12 software was adopted to analyze the data [17]. Furthermore, the DPABI toolbox was used to display the results [18]. The first 10 time points of the fMRI images were discarded due to the unstable magnetization in the early stage of the BOLD sequence. BOLD images were calibrated with slice time and motor function. The high-resolution images of the structure scan were co-registered with the functional data and then standardized by applying the Montreal Neurological Institute (MNI) T1 template. The BOLD images were then standardized using the parameters gained from the standardization of the structure images. Finally, images were smoothed with a Gaussian FWHM kernel of 6 mm.

### 2.8. General Linear Modeling

Model fit was determined with statistical time series analysis in the general linear model (GLM). Across each run, within-subject contrasts of two MI preparations and rest and two MIs and rest were calculated with a fixed-effects (first-level) analysis. The maps were thresholded using an initial threshold of *p* < 0.001 (uncalibrated). Only clusters at a significance threshold of *p* < 0.05 corrected for family-wise error (FEW) were reported. Anatomical locations and Brodmann regions were determined by utilizing the anatomy toolbox of DPABI.

### 2.9. Definition of Regions of Interest

Motor areas were the primary consideration as the most relevant brain areas for ME and MI tasks. It mainly contains the basic motor function areas of precentral gyrus (PreCG), postcentral gyrus (PoCG) and the advanced motor functions supplementary motor area (SMA). At the same time, the prefrontal cortex and supramarginal gyrus are associated with cognitive functions, including information recall and somatic stimuli [19]. We first focus on these brain regions and perform statistical analysis on the validated clusters, retaining the top five clusters with most voxels for each task condition as ROIs. We compare each motor task condition to its corresponding resting state to distinguish different movements. See Table 2 for activation area.

The results indicated that motor imaginations of the left- and right-hand activated the precentral gyrus (anterior gyrus) and paracentral lobule (the center of the left- and right-hand network connection). Additionally, the dorsolateral superior frontal gyrus dorsolateral part (SFGdl), which includes different lateral auxiliary motor regions, was also activated. Figure 3 shows activation in the somatosensory cortex of the postcentral gyrus and frontal cortex. These findings support the notion that, apart from the primary motor and premotor areas of the frontal lobe, the prefrontal lobe, responsible for cognitive thought processes, is also activated during motor imagination. These ROIs will be utilized for feature extraction.

### 2.10. Extraction of the Motor Image Features

In real-time data processing, SPM is used to individually register the localized ROIs with each subject. Thereby enabling the extraction of distinct brain signals. To counterbalance the global variations in the blood oxygen level dependence, the subsequent formula is employed, which calculates the average activation value within the ROI region. The activation value size is determined as the disparity between the average BOLD value in the target ROI and the average BOLD value in the resting ROI. This value is periodically updated once per volume acquisition time:(1)$${\mathrm{ACTIVE}=(\mathrm{BOLD}}_{\mathrm{test}}{-\mathrm{BOLD}}_{\mathrm{rest}}{)}_{\mathrm{targetROI}}$$
where ${\mathrm{BOLD}}_{\mathrm{test}}$ refers to the average value of ROI brain collected at the present time, and ${\mathrm{BOLD}}_{\mathrm{rest}}$ refers to the average value of ROI brain at rest.

fMRI reflects the change of blood oxygen concentration and has hemodynamic delay, which causes a delay in the detection of brain states. After approximately 2~6 s of active stimulation, the blood oxygen level in the brain functional area reaches its peak and transitions into a corresponding stable state. Therefore, during the screening of training data, the voxel activation level removes the transitional sequence between different functional states. According to statistical results, the activation level remains relatively stable after removing four volumes.

### 2.11. Dynamic Window-Level Granger Causality Model

Research has shown that GCM is an effective method for studying functional connectivity based on fMRI data. But this method of fMRI data processing is sensitive to the ROIs, so we selected seed nodes from the ROIs for Granger causality analysis. Based on the brainnetome atlas of 246 brain regions published by the Chinese Academy of Sciences [20], we extracted four seed nodes with the highest correlation in *t*-test of the ROIs: A1_2_3ulhf_l, A9_46v_l, A1_2_3ulhf_r, and A9_46v_r (Figure 4).

Then, we determined the coordinates of these seed nodes’ ROIs. The ROI radius for A9_46v_l and A9_46v_r was set to 5 mm, while the ROI radius for A1_2_3ulhf_l and A1_2_3ulhf_r was set to 10 mm. Taking Sub1 as an example, the Talairach coordinates (x, y, z in mm) for A1_2_3ulhf_l, A9_46v_l, A1_2_3ulhf_r, and A9_46v_r are (−29, −7, 64), (−48, 63, 28), (18, −3, 64), and (35, 67, 32), respectively.

The brain regions to which these nodes belong, in the brainnetome atlas, are associated with motor execution, finger movement, cognition, and memory functions. Then, based on temporal invariance, the extracted mean BOLD sequences from the seed nodes were subjected to DWGC analysis.

The basic principle of Granger causality model is to assume that there are two stationary time series:(2)$$\left(Y_{i1},Y_{i2},Y_{i3},\ldots ,Y_{it},\ldots )(Y_{j1},Y_{j2},Y_{j3},\ldots ,Y_{jt},\ldots \right)$$
where Granger causality defines $Y_{i}$ as the cause of $Y_{j}$, if the series i provides useful information when predicting the future values of series j:(3)$$E_{t}(g_{t}(Y_{j,t+1}|Y_{j,<t},Y_{i,<t}))\neq E_{t}(g_{t}(Y_{j,t+1}|Y_{j,<t}))$$
then we think that there may be a causal relationship between time series $Y_{i}$ and time series $Y_{j}$, and where $E_{t}$ is the accuracy expectation of the prediction function g of time series. It is important to note that Granger causality is still a statistical correlation, as it lacks the necessary causal identification.

It is necessary to determine the optimal range for the maximum window length for time-varying causal relationship values between brain regions. Aiming at finding the dynamic causality at the window level, we consider two forms of time series fitting on each sliding window (t,t+k− 1). After undergoing statistical analysis and iterative optimization, a window length of k = 4 is ultimately determined. This window length establishes the most significant correlation between the causal relationship of brain intervals and the experimental paradigm.

Finally, the nonlinear auto regressive model of time series is modeled on the dynamic window scale of k = 4, and the mean square error (MSE) is used as the $E_{t}$ measurement accuracy:(4)$$L_{1}=E(MSE({\overset{⌢}{Y}}_{i,t\sim t+k-1},Y_{i,t\sim t+k-1}|Y_{i,<t}))$$
(5)$$L_{2}=E(MSE({\overset{⌢}{Y}}_{i,t\sim t+k-1},Y_{i,t\sim t+k-1}|Y_{i,<t},Y_{j,<t}))$$
where $L_{1}$ refers to the estimation accuracy of prediction using the sequence $Y_{i}$, $L_{2}$ refers to the estimation accuracy of joint prediction using the sequences $Y_{i}$ and $Y_{j}$, and ${\hat{Y}}_{i,t\sim t+k-1}$ depicts prediction on the sliding window (t,t+k− 1). We determined the existence of causality by setting a reasonable threshold ω based on the value of $F_{statistic}$:(6)$$F_{statistic}=\frac{L_{1}}{L_{2}}$$

Finally, set a reasonable threshold to further constrain the causal relationship:(7)$$F=\left\{\begin{array}{c} F_{statistic},F_{statistic}\geq \omega \\ 0,\;F_{statistic},<\omega \end{array}\right.$$

The final F values between each node will serve as the input features.

## 3. Prediction Model

In order to accurately and efficiently identify the state changes in rest and MI, as well as classify left- and right-hand motion imagination, we conducted experiments using four different algorithms. To ensure comprehensive testing, we also modified certain parameters of the algorithms. The specific algorithms used are GaussianNB, SVM, XGBoost, and lightGBM. For more details on the parameter changes, refer to Table 3.

GaussianNB is a classical machine learning algorithm and one of the few classification algorithms based on probability theory. Its fundamental principle involves computing the conditional probability of each feature for different classes using a known sample dataset. It then calculates the posterior probability of the samples to be classified by applying Bayesian theorem. The class with the highest posterior probability is considered the class to which the samples belong [21].

SVM, on the other hand, performs non-linear mapping of the data to higher dimensional spaces. It constructs a hyperplane to effectively separate the different classes. The choice of the boundary function is often adjusted to determine the most suitable fit for a given dataset.

XGBoost, an implementation of the boosting algorithm, employs multiple base learners to learn the differences between model values and actual values. It boasts excellent performance, simplicity, and high speed, making it well suited for real-time classification needs [22].

LightGBM draws inspiration from the gradient boosting machine (GBM) model, and it can be viewed as an evolutionary version of gradient boosting decision tree. Its primary concept involves iteratively training an optimal model using weak classifiers. This approach yields good training effects without the risk of overfitting [23].

To acquire statistical results, each classifier has undergone multiple training sessions. To assess the classifier’s performance, we evaluate the accuracy and Kappa statistic by utilizing the confusion matrix. Accuracy measures the percentage of correctly classified instances, while Kappa statistics represent coefficients of consistency. These statistics gauge the correlation between the expected and achieved results. The confusion matrix, on the other hand, offers insights into the rates of false positives, false negatives, true positives, and true negatives.

All tests utilize K-fold cross-validation as the testing method. In this approach, the algorithm partitions the dataset into K subgroups. The classifiers are subsequently trained individually, while the remaining subgroups are employed to construct a test set. The overall performance is then calculated as the average of K tests.

## 4. Related Works

To provide a more accurate and straightforward demonstration of the effectiveness of the dynamic Granger causal model in extracting brain interval connections, the results are presented independently. After optimizing the algorithm, the BOLD values of the four extracted nodes are computed using DWGC with a window length of four. The dynamic relationships of effective connections between the target seed nodes are depicted in Figure 5.

The results demonstrate significant changes in the effective connections between the ROIs when calculated with DWGC (*k* = 4). Moreover, the causal values of the two prefrontal nodes to the anterior gyrus differ noticeably in simultaneous measurements. The results show that there are significant disparities in motor imagination between the left and right hand, specifically between resting state and task state. Such distinctive features present promising results for real-time classification of left-hand and right-hand motion imaginations.

## 5. Results and Discussion

### 5.1. Group Analysis

For selecting the subjects, firstly, subjects who did not exhibit any finger or hand movements throughout the entire research process were chosen. During the experiment, eye-tracking data was continuously monitored using an Eyelink S1000 system. EyeLink log files provide horizontal position, vertical gaze position, pupil size measures for each timepoint, and tags corresponding to blink onset and offset. We believe that the subjects who had missing or abnormal eye-tracking data indicated an incomplete completion of the experiment. And based on the interviews conducted after the experiment, we excluded subjects who exhibited signs of fatigue during the experiment. Finally, three subjects who meet the experimental requirements were selected. A study conducted by Amar et al. demonstrates that conducting motor imagery experiments with small sample sizes can still yield significant and accurate results when applied to large datasets [24,25].

Baseline conditions were established before the experiment. To confirm the establishment of similar activation levels before the MI task in rtfMRI experiment, we examined any differences in activation among the subjects. Analysis (based on paired *t*-tests) revealed no significant differences in brain activity before the experiment. The threshold range of 0.01 < *p* < 0.05 did not alter our observation results.

### 5.2. Results of the Classification

Table 4 shows the performance of classifiers with different parameters in the left–right hand binary classification task.

Statistical results indicate that all the classification algorithms exhibit high classification accuracy and Kappa value. Among them, the RBF kernel SVM classifier achieved the highest classification accuracy and Kappa coefficient with an optimal parameter combination of C = 19 and gamma = scale. From a mathematical perspective, the Gaussian kernel function introduces nonlinear mapping, which can better adapt to the complex distribution of data compared to linear kernel. Furthermore, utilizing the gamma value, which is automatically calculated by standard deviation, can adjust the varying ranges of different features, consequently, enhancing the model’s generalization ability and accuracy. Overall, this outcome aligns with our expectations as the cortical mapping distance indicates significant spatial separation of motion-related brain activation areas [26], and the activation patterns also exhibit noticeable differences, making them easy to distinguish.

And we compared the effectiveness of feature extraction based on DWGC and found that the effective connection features improved the classification accuracy across various classifiers (Figure 6). In order to demonstrate the significance of DWGC in improving classification performance, a paired-sample *t*-test was conducted on two sets of classification results. The results indicated that among the five classifiers used, all exhibited a statistically significant increase in accuracy with the implementation of DWGC (*p* < 0.05).

In the experimental task, we also studied the timing of information acquisition during motion imagination. During rt-fMRI experiment, the block of a MI task lasted for 10 s. The decoding accuracy rate is calculated separately for each TR, representing the information decoding time. Specifically, the trained models are inputted to the sample features individually at each moment in the block. This process is repeated for all samples to generate a decoding accuracy (DA) curve (Figure 7).

Each DA curve consists of five data points. The figure displays the DA curves for the effective connection features extracted with DWGC model and those without them. Each data point on the DA curve represents the average DA value, expressed as a percentage, across all three subjects during the task. The red dotted line represents the theoretical estimate of the random decoding accuracy level (50%). The DA values for all MI tasks were slightly higher than the random level (50%) in volume 1, and they gradually increased, peaked at volume 3 or 4, and then stabilized. After statistical analysis, the average accuracy of DA time curve with DWGC effective connection is higher than that of DA value time curve without DWGC feature.

Figure 8 shows the accuracy of three classifications among three different subjects including left-hand (LH) imagery, right-hand (RH) imagery and rest. The average accuracy of classification of three subjects is 65.6%.

The results showed that the classification accuracy of subjects’ right-hand MI was higher. One possible reason is that all subjects are right-handed, so when they perform the task of imagining right-hand movements, and there is a more noticeable active stimulation of the brain area, resulting in a higher level of activation in the relevant brain regions. From the above analysis, it can be seen that there are certain differences in the network measures of the motor imagination brain network among different subjects under the same experimental environment and process. This indicates that different subjects show differences in their ability to perform MI tasks.

The workstation utilized in this paper is equipped with an i7-10750H CPU clocked at 2.6 GHz. Based on Open-NFT platform, the average processing time of each TR is 1.15 s, which is less than the real-time classification requirements of a brain–computer interface, given a TR scanning time of 2 s.

## 6. Conclusions

This study proposes a method that combines dynamic Granger causality analysis to improve real-time classification of left- and right-hand motor imagery. Based on the analysis of brain region activation, effective connectivity features between Frontal_Sup_Medial and Postcentral gyrus are incorporated, resulting in a 3% increase in accuracy compared to traditional methods based solely on brain region activation values. The results demonstrate a strong correlation between the extracted effective connectivity features and the MI task, with the classification accuracy peaking within the third TR value after the start of the task. To our knowledge, this study is the first to propose and apply DWGC in the field of rt-fMRI-based BCI. Compared to brain region classification based on BOLD activation values, this method effectively improves the accuracy of state detection.

In neurofeedback techniques, providing effective, accurate, and rapid feedback is crucial for patients with movement disorders. Compared to other techniques such as EEG, fMRI neurofeedback offers a more biologically plausible and interpretable approach [27]. This study provides a reliable technical method using fMRI technology for the treatment of patients, offering a new training methodology for MI-BCI rehabilitation training. Furthermore, the discovery of effective connectivity between the PoCG and MFG helps unravel the neural circuits and mechanisms associated with motor control in the brain, which holds significant implications for cognitive neuroscience and the field of motor control.

Our future research will focus on decoding complex motion imagination within a larger brain network, which will have broader practical applications.

## Figures and Tables

**Figure 1 brainsci-13-01406-f001:**
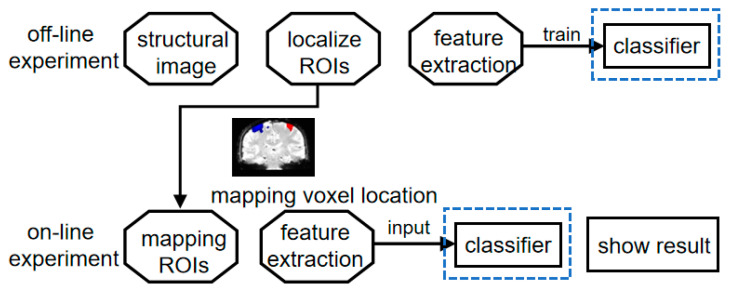
Off-line experiment involves motor imagination tasks for the left and right hand. The objective of the experiment is to identify the activated functional brain regions in the subjects. Subsequently, features are extracted offline from the data and used to train the classifiers for left-hand and right-hand movements. In on-line experiment, the functional brain regions identified in off-line experiment were mapped in the subjects’ brains. The classifiers trained using off-line experiment data were then employed to classify the subjects in real time and display the classification results immediately.

**Figure 2 brainsci-13-01406-f002:**
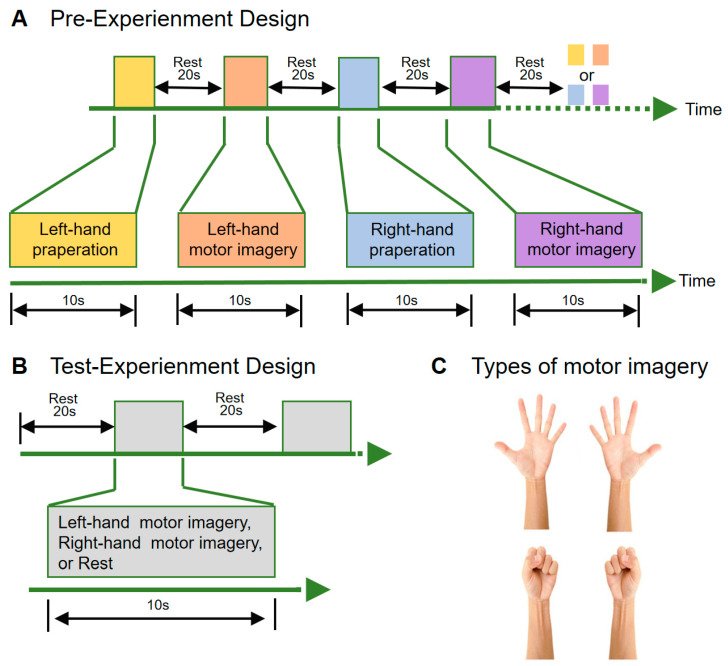
Tasks and experimental paradigm. (**A**) Pre-experimental paradigm, as depicted in Figure **A**, involved four groups of tasks conducted within a single run. These tasks involved left-hand imagery preparation, left-hand motor imagery, right-hand imagery preparation, and right-hand motor imagery. Each task lasted 10 s with a 20 s rest between tasks. The tasks are performed randomly, and the experiment lasts for a total of 300 s. (**B**) Test-experiment paradigm is shown in the figure, the subjects performed left-hand motor imagination or right-hand motor imagination or rest for 10 s and rested for 20 s. The experiment lasted for 300 s. (**C**) The hand movements of the participants required for motor imagery.

**Figure 3 brainsci-13-01406-f003:**
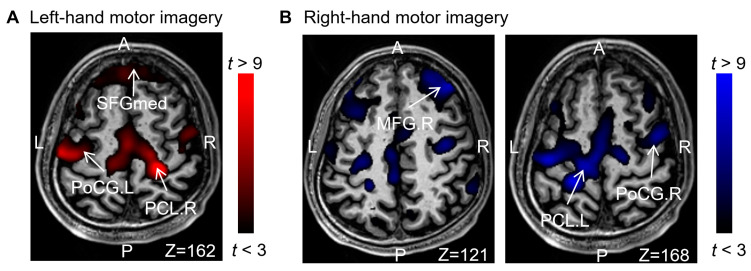
Comparison of activation areas for all tasks: (**A**) left motor imagery > rest and (**B**) right motor imagery > rest.

**Figure 4 brainsci-13-01406-f004:**
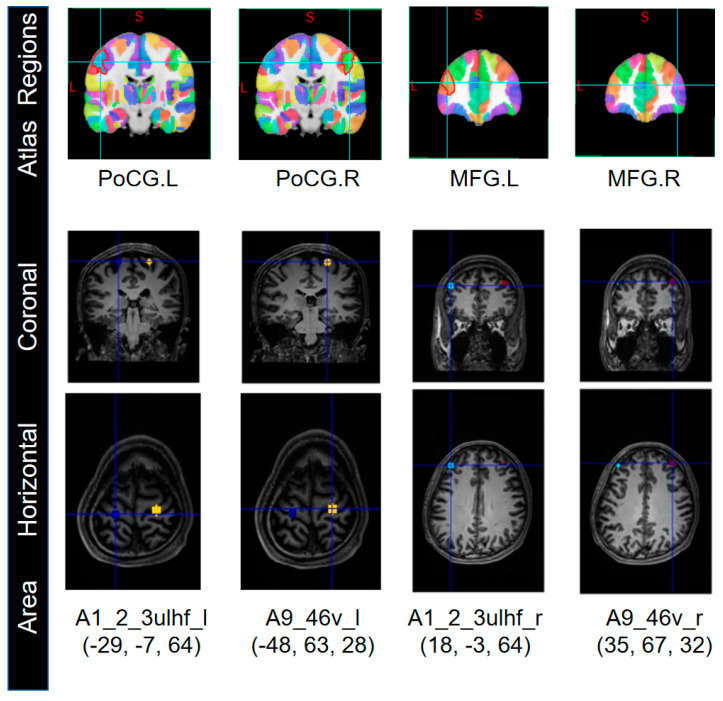
The selected four seed nodes correspond to the brainnetome atlas (Sub 1).

**Figure 5 brainsci-13-01406-f005:**
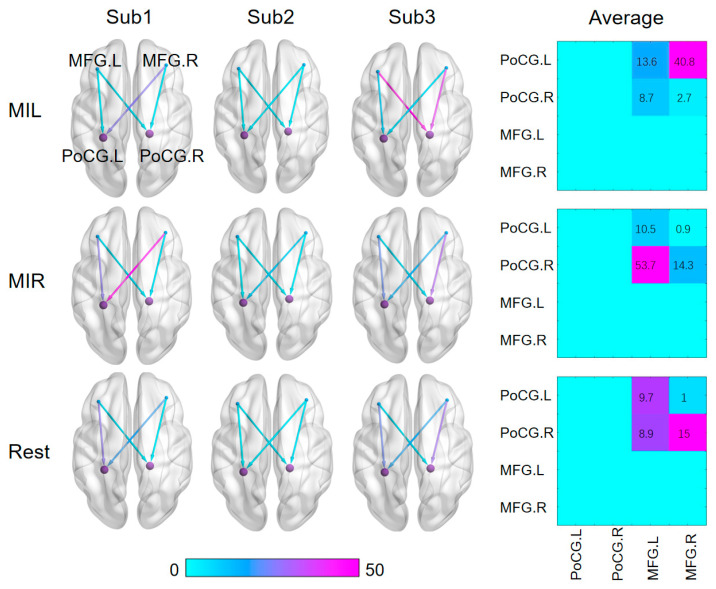
The figure shows visualization results of the statistical analysis based on the Granger causality between the seed nodes in different brain states for three subjects. And the average effective connectivity matrix was calculated from middle frontal gyrus (MFG) to PoCG.

**Figure 6 brainsci-13-01406-f006:**
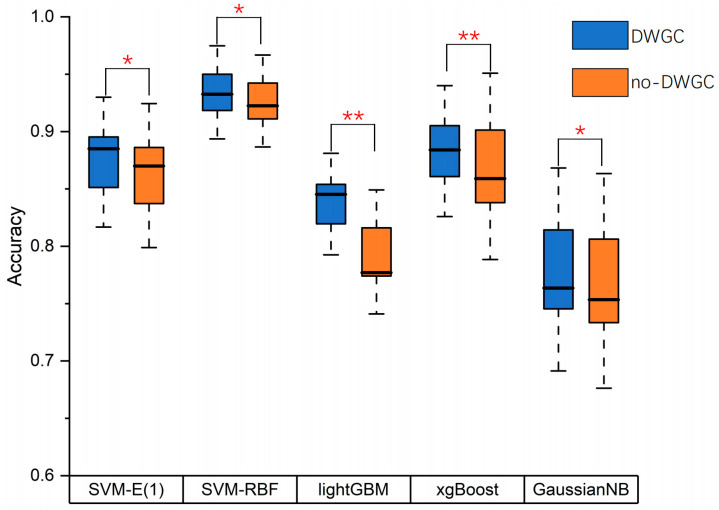
Accuracy results of binary classifications with different classifiers. * = *p* < 0.05, ** = *p* < 0.01.

**Figure 7 brainsci-13-01406-f007:**
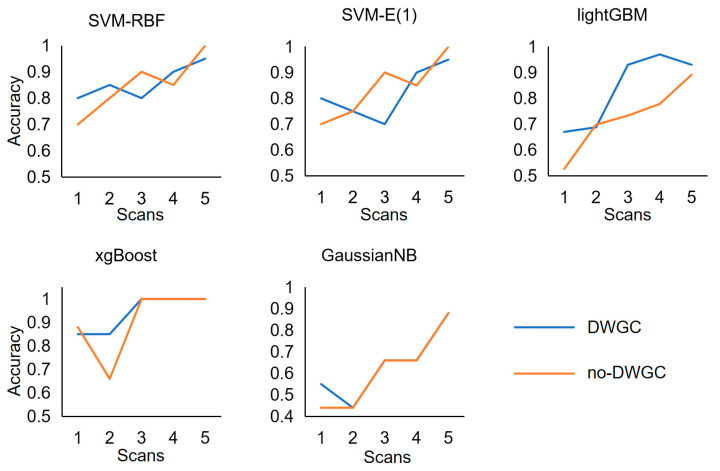
Decoding correct rate time based on DWGC and BOLD in block unit.

**Figure 8 brainsci-13-01406-f008:**
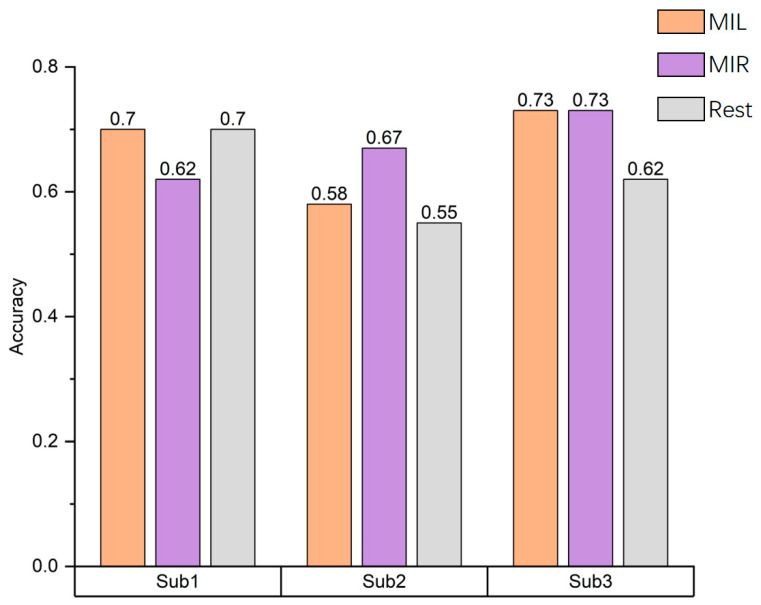
Accuracy results of three different subject classifications. The decoding results reveal varying classification accuracy among different subjects, with real-time decoding accuracy surpassing the theoretical estimation of random level by 35%.

**Table 1 brainsci-13-01406-t001:** Magnetic resonance scanning parameters.

Key Parameters	Functional Image Acquisition Parameters	Structural Image Acquisition Parameters
TR	2000 ms	2300 ms
TE	30 ms	2.26 ms
slices	62	192
slice thickness	2 mm	1 mm
FOV	192 × 192 mm^2^	256 × 256 mm^2^
flip angle	90°	8°
matrix size	112 × 112	256 × 256
voxel size	2 × 2 × 2 mm^3^	1 × 1 × 1 mm^3^

Note. TR, repetition time; TE, echo time; and FOV, field of view.

**Table 2 brainsci-13-01406-t002:** Motor imagination task activation area.

Contrast	Brain Area	BA	Side	Cluster Size	Peak MNI Coordinate		
					X	Y	Z
MIL	SMG	40	L	238	−53	−50	49
	SPG	7	R	43	31	−56	66
		7	L	42	−22	−59	65
	SSC	2	R	26	46	−40	48
	DLPFC	46	R	56	36	45	26
	dPCC	30	R	30	16	−36	−13
	PreCG	6	R	6	24	−20	74
MPL	SMG	40	L	273	−50	−36	50
	SMA	6	R	36	6	20	68
	SPG	7	R	418	10	−60	68
	DLPFC	9	R	59	54	12	36
		9	L	33	−38	38	32
	APFC	10	R	35	46	48	4
MIR	PreCG	6	L	123	−31	−23	69
	SSC	2	L	102	−32	−43	62
	SAA	5	L	64	−10	−51	67
	SMG	40	L	62	−32	−40	49
	PreCG	4	L	36	−54	−14	57
	SSC	1	L	12	−27	−42	71
MPR	SSC	1	L	193	−30	−41	73
	APFC	10	L	308	−4	64	2
	MI	4	R	8	46	−16	50
	PreCG	6	L	36	−18	−14	64
	SSC	2	L	19	−30	−38	74

Note. Coordinates are at the MNI space. FDR q value = 0.05; spatial extent k > 5 voxels; BA, Brodmann’s area; MIL, left-hand grasping motor imagination; MPL, left-hand grasping motor preparation; MIR, right-hand grasping motor imagination; MPR, right-hand grasping motor preparation; L, left; R, right; PreCG, precentral gyrus; SMA, supplementary motor area; SMG, supramarginal gyrus; SPG, superior parietal gyrus; SSC, somatosensory cortex; DLPFC, dorsolateral prefrontal cortex; dPCC, dorsal posteriorcingulate cortex; APFC, anterior prefrontal cortex; SAA, somatosensory association cortex, MI, motor imagery.

**Table 3 brainsci-13-01406-t003:** Classifier training parameters.

Classify Module	Parameters
GaussianNB	
SVM	Polynomial kernel:exponent (E) = 1
	RBF kernel
LightGBM	Max_depth = 5, learning_rate = 0.8 n_estimators = 1000
xgBoost	Max_depth = 8, learning_rate = 0.08 n_estimators = 1000

**Table 4 brainsci-13-01406-t004:** Average accuracy and variance of classifier.

Classifier	Parameter	Accuracy	Kappa Statistic
		Mean	Mean
GaussianNB		77.7%	55%
SVM	Polynomial kernel:exponent (E) = 1	91%	51.02%
	RBF kernel	94.25%	65.59%
LightGBM	Max_depth = 5 learning_rate = 0.8 n_estimators = 1000	80.93%	44.14%
xgBoost	Max_depth = 8 learning_rate = 0.08 n_estimators = 1000	91.1%	47.71%

## Data Availability

The data that support the findings of this study are available from the corresponding author upon reasonable request.

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
