# Peer review of "Real-Time Classification of Motor Imagery Using Dynamic Window-Level Granger Causality Analysis of fMRI Data"

_brainsci, 2023, doi:10.3390/brainsci13101406_

Round 1

Reviewer 1 Report

The paper is interesting and require some modifications as follows:

-in the abstract, please explain where data comes from? and how many subjects were used for processing.

-In the abstract, it is necessary to mention one of the best achieved accuracy level.

-in the abstract, please introduce the abbreviations which are repeated in the text more than a once. for example, MVPA is introduced in the abstract, but it did not used. please remove the abbreviations.

 -please introduce the abbreviations in the same format. some are uppercase and some are lowercase, for example, MI is uppercase and svm is lowercase.

 -please use the keywords which are employed in the abstract. for example, svm is not used in the abstract whilst it is in the keyword.

 -the keywords required updating by using the complete word combinations and the abbreviations.

- the grammatical and editorial corrections are required. for example, In the introduction, please remove ‘’-‘’ in the brain-computer

 -in the introduction, the SVM is not introduced.

 -please do not leave any of the sections empty. for example, in the ‘’ 2. Materials and Methods’’ it is expected to explain in a few sentences what is the content in the next subsections.

 -As it is a BCI study, it would be more beneficial if use the last review papers in the BCI applications in the introduction such as

 Hekmatmanesh A, Nardelli PH, Handroos H. Review of the state-of-the-art of brain-controlled vehicles. IEEE Access. 2021 Jul 27;9:110173-93.

-please explain if authors considered the subject’s health history, medicine history and healthiness.

 -in the ethical statement please mention if the subjects signed the letter of consent for participating the experiment.

 -please introduce the abbreviations in all Tables. for example, FOV, TR, TE. Also please select a title for the first column.

- there is a large number of abbreviations, it is recommended to open a table named NOMENCLATURE and present all them here.

 - please remove the indents after the equations which are used for the parameters explanations. for example, ‘’BOLDtest refers to…’’ this sentence is introducing the parameters of in equation 1. the sentence is not finished. Also it is recommended to use comma after equation, then introduce the parameters in the equation (without indent). This mistake repeated for all the equations.

 -       in the discussion part, it would be supportive to explain mathematically, which parameters or mathematical combinations of the algorithm cause higher accuracy result.

The English is understandable but more editorial work is still required to be eligible for publication.

Author Response

please refer the attachment

Reviewer 2 Report

The paper discusses the application of a DWGC model for extracting brain connectivity patterns in the context of brain-computer interaction during motor imagery tasks. The authors conducted experiments involving left and right hand motor imagery and analyzed the functional connectivity between various brain regions. They also utilized machine learning classifiers to decode motor imagery states in real-time based on extracted features.

The paper presents promising findings in terms of decoding accuracy for left and right hand motor imagery tasks. The DWGC model showed improved response speed and classification accuracy, making it a potential candidate for real-time BCI applications.

However, there are several aspects that could be improved in the paper.

1.      The introduction tries to give a comprehensive view of research on motor imagery. However the citations from recent research are missing, making the information presented not comprehensive and slightly outdated. I suggest adding more citations from recent research as well. For example, since the study is about hand grasping it might be useful for them to look into studies by Lakshminarayanan et al. about motor imagery for hand tasks.

·        Lakshminarayanan, K., Shah, R., Daulat, S. R., Moodley, V., Yao, Y., & Madathil, D. (2023). The effect of combining action observation in virtual reality with kinesthetic motor imagery on cortical activity. Frontiers in Neuroscience, 17, 1201865.

·        Lakshminarayanan, K., Shah, R., Yao, Y., & Madathil, D. (2023). The effects of subthreshold vibratory noise on cortical activity during motor imagery. Motor Control, 1(aop), 1-14.

2.      The study is based on a small sample size of only six subjects. It's essential to address the limitations associated with this small sample and discuss the implications for the generalizability of the results to a broader population. To this end, the method section could be improved, by adding some recent MI studies that have used similar sample sizes in different backgrounds, and have successfully addressed these concerns; please refer the following references:

·        Di Flumeri et al. “Brain–Computer Interface-Based Adaptive Automation to Prevent Out-Of-The-Loop Phenomenon in Air Traffic Controllers Dealing With Highly Automated Systems”

·        Gomez et al. “User Engagement Comparison between Advergames and Traditional Advertising Using EEG: Does the User’s Engagement Influence Purchase Intention?”

The authors are welcome to include these citations and more from other researchers to give a comprehensive view on the field.

3.      Line 45 – Change title to “Apparatus”

4.      While the paper mentions statistical analysis and significance (confusion matrix), it lacks specific details about the statistical tests used and their results. To strengthen the paper's credibility, the authors should provide more information on the statistical methods employed and the significance of their findings.

5.      While the paper discusses the technical aspects of the research, it could benefit from a more detailed discussion section with citations of the practical applications and implications of the findings. How might this research impact the field of neurorehabilitation or brain-computer interfaces for individuals with motor disabilities? The authors should provide a clearer discussion of the potential real-world applications of their work.

Author Response

please refer the attachment

Round 2

Reviewer 2 Report

Thank you for diligently addressing all the comments and making appropriate changes to the manuscript.